# Length of Hospital Stay for Osteoarthritic Primary Hip and Knee Replacement Surgeries in New Zealand

**DOI:** 10.3390/ijerph16234789

**Published:** 2019-11-29

**Authors:** Chunhuan Lao, David Lees, Sandeep Patel, Douglas White, Ross Lawrenson

**Affiliations:** 1Waikato Medical Research Centre, The University of Waikato, Hamilton 3240, New Zealand; ross.Lawrenson@waikatodhb.health.nz; 2Orthopaedic Department, Tauranga Hospital, Tauranga 3143, New Zealand; dlee078@gmail.com; 3Orthopaedic Department, Waikato Hospital, Hamilton 3240, New Zealand; sandeep.patel@waikatodhb.health.nz; 4Rheumatology Department, Waikato Hospital, Hamilton 3240, New Zealand; douglas.white@waikatodhb.health.nz; 5Strategy and Funding, Waikato District Health Board, Hamilton 3240, New Zealand

**Keywords:** hip replacement, knee replacement, osteoarthritis, length of stay

## Abstract

This study aims to explore the length of stay (LOS) of publicly funded osteoarthritic primary hip and knee replacement surgeries in New Zealand. Patients with osteoarthritis who underwent publicly funded primary hip and knee replacement surgery in 2005–2017 were included. We have identified 53,439 osteoarthritic primary hip replacements and 50,072 osteoarthritic primary knee replacements. LOS has been reduced by almost 40% over the last 13 years. Logistic regression showed that women, Māori, Pacific and Asian patients, older patients, people with more comorbidities and those having opiates on discharge and patients in earlier years were more likely to have extended LOS following hip replacements and knee replacements. Regional differences were noted in LOS between the Waitemata District Health Board (DHB) compared to Tairāwhiti DHB where patients were the most likely to have a LOS of more than 5 days after hip and knee replacements. LOS after hip and knee replacements has been reduced dramatically. Women, Māori, Pacific and Asian patients, older patients and people with more comorbidities are more likely to have extended LOS. Patients dispensed opiates on discharge had a longer LOS. There are great geographical variations in LOS for primary hip and knee surgeries in New Zealand.

## 1. Introduction

Osteoarthritis of the hip and knee is one of the most common causes of reduced mobility [1]. In New Zealand, osteoarthritis affects 10.6% of adults [2]. Hip and knee replacements for osteoarthritis can help alleviate pain and improve function. The New Zealand Joint Registry reports that osteoarthritis was responsible for 87% of primary hip arthroplasties and 84% of primary knee arthroplasties [3]. The number of hip and knee replacement surgeries in New Zealand has been increasing over time. There were 8785 primary hip replacement surgeries and 7765 primary knee replacement surgeries performed in 2016, compared to 4114 and 2429 in 1999, which poses an increasing burden to the healthcare system [3].

Hospital inpatient costs of arthritis were estimated to be NZ$321 million in 2018, and they were dominated by osteoarthritic hip and knee surgeries [4]. Length of stay (LOS) for surgery plays an important role in the costs and the cost-effectiveness of hip and knee surgeries [5,6,7,8]. Reduced LOS following joint replacement surgery is a generalized trend around the world [7,9]. Decreased LOS is an indication of reduced costs and increased cost-effectiveness. In a UK study, the LOS dropped from 16.0 days in 1997 to 5.4 in 2014 for primary knee replacement, and from 14.4 to 5.6 for primary hip replacement, leading to savings of £1537 and £1412 per patient (without adjusting for inflation) over the 23 years [7]. A US study reported a cost saving of US$3245 per patient for primary total knee arthroplasty after shortening the LOS by 0.7 days [8]. The public healthcare resources are limited and reduction of costs in joint replacement surgeries can lead to more resources for saving lives and improving quality of life in other health domains.

The New Zealand health care system is a mix of government funded and private health care. Publicly funded primary hip replacements accounted for 54% of the total number of all primary hip replacements in 2005–2016, and publicly funded primary knee replacements accounted for 59% of the total numbers of all primary hip replacements [3]. This study aims to explore the LOS of publicly funded osteoarthritic hip and knee replacement surgeries in New Zealand and to examine the factors that influence LOS.

## 2. Methods

This study included patients with osteoarthritis who underwent publicly funded primary hip and knee replacement surgery in 2005–2017 [10]. These records were identified from the National Minimum Dataset (NMD) that stores all publicly funded hospital inpatient and day-patient discharge information nationally. The ICD-10-AM ACHI Procedure Codes (Version 3) were used to extract the primary hip and knee replacement surgeries. These records were cross referenced with the New Zealand Joint Registry [3] data to exclude the admissions not for primary hip or knee replacement surgeries. The NMD also records patient’s diagnosis for the admission and other comorbidities. Patients without a diagnosis of osteoarthritis were excluded. Comorbidities for calculating the Charlson comorbidity index score were identified from the diagnostic codes based on the coding algorithms developed by Quan et al. [11].

The PHARMS dataset was linked to the NMD through patients’ National Health Index (NHI) numbers to identify the use of strong opiates after surgeries [12]. The NHI number is a unique identifier that is assigned to every person who uses health and disability support services in New Zealand. The PHARMS dataset stores all publicly funded pharmaceutical dispensing records. The strong opiates included in this study were dihydrocodeine tartrate, fentanyl citrate, fentanyl, methadone hydrochloride, morphine hydrochloride, morphine sulphate, morphine tartrate, oxycodone pectinate, oxycodone hydrochloride and pethidine hydrochloride.

We estimated the average LOS after surgery by gender, ethnicity (Māori, Pacific, Asian and European/others), age group (<40, 40–49, 50–59, 60–69, 70–79 and 80+ years), Charlson comorbidity index score (0, 1, 2 and 3+), year of surgery, use of opiate or not after surgery and District Health Board (DHB) where the surgery was performed. There are 20 DHBs in New Zealand. They are responsible for providing or funding the provision of health services in their district. The public hospitals where these joint replacement procedures are performed are owned and funded by DHBs. An independent-samples *t*-test and one-way ANOVA were used to compare the average LOS between subgroups. We used two definitions of extended LOS: (1) over 5 days and (2) over 7 days. A logistic regression model was used to examine the impact of all these factors on extended LOS and estimate the odds ratios after adjustment for age, gender, ethnicity, Charlson comorbidity index score, year of surgery, use of opiate or not after surgery and DHB. All data cleaning and analyses were performed in R 3.5.0 ( R Foundation For Statistical Computing, Vienna, Austria).

## 3. Results

We identified 53,439 publicly funded osteoarthritic primary hip replacement surgeries and 50,072 osteoarthritic primary knee replacements in 2005–2017 (Table 1). The numbers of hip and knee replacement surgeries have been increasing over time. Around 93% of primary hip replacements and 98% of primary knee replacements were performed in patients aged 50+ years. Slightly more hip and knee replacement surgeries were performed in women (54.3% hips and 53.3% knees) than in men. Māori were more likely to have primary hip replacement surgeries than knee replacement surgeries, but Asian and Pacific people were more likely to have knee replacement surgeries than hip replacement surgeries. The majority of patients had no severe comorbidities: 87.6% for hip surgeries and 85.1% for knee surgeries. Twenty-two percent of patients undergoing hip replacement surgeries and 30.5% of knee replacement patients had strong opiate drugs post operation.

Women had a longer LOS than men (5.3 vs. 4.7 days for hip surgeries and 5.4 vs. 5.1 after knee surgeries; Table 1). The LOS generally increased with age except for the age group of less than 40 years old. Māori patients had the shortest LOS than other ethnic groups. However, when we compared the LOS between ethnic groups after stratifying them by age group (Table 2), Māori and Pacific patients generally had longer LOS than Europeans and others by age group.

The LOS increased with Charlson score: from 4.8 days for score 0 to 8.3 days for score 3+ for hip surgeries, and from 5.1 days for score 0 to 7.9 days for score 3+ for knee surgeries (Table 1). Patients who had opiates after surgery had a longer LOS than those did not have opiates. The overall LOS has been decreasing over time: from 6.2 days in 2005 to 3.8 days in 2017 for hip surgeries, and from 6.6 days in 2005 to 4.1 days in 2017 for knee surgeries.

South Canterbury DHB (4.3 days) had the shortest LOS after hip surgeries (Table 3), followed by Nelson–Marlborough DHB (4.4 days) and Capital and Coast DHB (4.6 days). Tairāwhiti DHB (5.9 days), Auckland DHB (5.6 days) and Waikato DHB (5.5 days) had the longest LOS after hip surgeries. Nelson–Marlborough DHB (4.4 days) had the shortest LOS after knee surgeries, followed by Capital and Coast DHB (4.8 days) and South Canterbury DHB (4.8 days). Auckland DHB (6.0 days), Wairarapa DHB (5.9 days) and Tairāwhiti DHB (5.7 days) had the longest LOS after knee surgeries.

For primary hip replacement, 29.9% of patients had a LOS of more than 5 days and 10.5% had a LOS of more than one week. For primary knee replacement, 34.2% of patients had a LOS of more than 5 days and 12.4% had a LOS of more than one week. Logistic regression (Table 4 and Table 5) showed that after adjustment, women, Māori, Pacific and Asian patients, older patients, people with more comorbidities, those having opiates post operation and patients in earlier years were more likely to have extended LOS in terms of both hip surgeries and knee surgeries. Compared to Waitemata DHB, patients in Tairāwhiti DHB were the most likely to have a LOS of more than 5 days after hip and knee surgeries (odds ratio: 4.88 (95% CI: 4.09–5.81) and 2.78 (95% CI: 2.30–3.37) and the most likely to have a LOS of more than one week after hip surgeries (odds ratio: 3.13 (95% CI: 2.53–3.86)). Compared to Waitemata DHB, patients in Auckland DHB were the most likely to have a LOS of more than 7 days after knee surgeries (odds ratio: 1.79 (95% CI: 1.58–2.03). For hip surgeries, South Canterbury DHB was the least likely to have an extended LOS of more than 5 days (0.75 (95% CI: 0.63–0.88)), and of more than 7 days (0.46 (95% CI: 0.35–0.61)). For knee surgeries, Nelson–Marlborough DHB was the least likely to have an extended LOS of more than 5 days (0.56 (95% CI: 0.49–0.64)), and of more than 7 days (0.45 (95% CI: 0.36–0.55)). 

## 4. Discussions

The average LOS was reduced by 2.4–2.5 days for primary hip and knee replacement surgeries over the 13 years period (from 2005 to 2017), leading to a cost reduction of approximately NZ$4000 per surgical case that is 13%–20% of the total cost [13,14]. The total cost savings for all the publicly funded primary hip and knee replacement surgeries in 2017 were approximately 42 million New Zealand dollars due to the reduced LOS.

Overseas studies demonstrated the same time trend of LOS following total hip and knee replacement [7,9]. The average LOS in the UK in 2014 was 5.6 days for primary hip replacement and 5.4 days for primary knee replacement, which were both slightly longer than in NZ: 4.4 days and 4.6 days respectively. However, the LOS in the US is even shorter: 2.75 days for hip replacement and 2.95 days for knee replacement in 2013 [9].

Improved processes have been shown to reduce LOS [7]. Protocols with an emphasis on recovery and rehabilitation have been introduced in elective orthopedics, including ‘Accelerated Rehabilitation’, ‘Fast-Track’, ‘Clinical Pathways’ and “Enhanced Recovery After Surgery (ERAS)” programs [15,16,17]. Many of these programs have been introduced to New Zealand and have contributed to the reduced LOS [18,19]. The LOS for hip and knee replacements could reduce further, with outpatient hip and knee replacement surgeries being possible. A US study found that 1-day LOS discharge after total hip and knee replacement is achievable and did not increase readmissions compared to 2-day LOS discharge [20].

A key component of every enhanced recovery protocol is the use of multimodal (opioid-sparing) analgesia involving a combination of different opioid and non-opioid analgesic and non-analgesic drugs [21]. Multimodal analgesia can reduce postoperative side effects and therefore facilitate the recovery process after surgery [22]. This study showed that patients who used opiates after hip and knee surgery had a longer LOS. This may represent higher levels of post-operative pain as well as the higher levels of inpatient opiate use influencing LOS.

Age at surgery, comorbidities, low income, obesity and female gender have been reported to be associated with increased LOS [5,6,7,23,24,25]. In our study, women had a longer LOS than men, which was consistent with other studies [6,26,27]. This could be attributed to differences in ways in which men and women respond to the disease, anesthesia and the surgery or to bias on the part of healthcare workers [6,26,27]. As expected, the LOS increased with age, because older patients have more comorbidities and have slower recovery after surgery. It has also been found that older patients were more likely to have experienced post-operative complications, admission to the intensive care unit (ICU) and be discharged to a skilled care facility [28]. Asian, Pacific and Māori patients were more likely to have an extended LOS of more than 5 days and 7 days than European/others after adjustment for other factors. This may be also related to differences in comorbidities in these groups. In this cohort, 27% of Asian patients, 26% of Pacific patients and 20% of Māori patients had at least one comorbidity, compared to 12% European/others. Other reasons may include the prevalence of obesity in Māori and Pacific patients [29]. Obesity is associated with a longer LOS after joint replacements [30].

There are great geographical variations in LOS after primary hip and knee surgeries in New Zealand. This may be partially explained by the differences in medical centers and the populations in different DHBs. Expert orthopaedic surgeons in big DHBs may have a shorter length of operation than generalist orthopaedic surgeons in small DHBs. Length of operation has been shown to increase the risk of a major complication, resulting in an increased LOS [31]. The Auckland DHB had a high Pacific population (10.3% vs. 6.5% for the national population), and the Tairāwhiti and Waikato DHB had a high Māori population (50.3% and 22.8% vs. 15.7% for the national population) [32]. The Nelson–Marlborough and South Canterbury DHB have an older and rural population with lower Māori and Pacific populations [32].

The strength of this study was that it was based on national datasets including over 100,000 primary hip and knee replacement surgeries. These datasets collect comprehensive data on patient characteristics, comorbidities and post discharge pharmaceutical dispensing [10]. One weakness is that this study did not include certain clinical information, e.g., pre- and postoperative hemoglobin levels and blood transfusions, which have been found to influence LOS [27]. Another limitation is that some peripheral centers without specialized services such as an ICU or access to dialysis transfer their high risk and dialysis patients to a tertiary centre for their surgery or post-op management, which is likely to influence the LOS but we could not identify these cases. Similarly, we could not identify whether some hospitals had skilled nursing facilities or extended rehabilitation facilities, which would affect the LOS. We had no data on social determinants such as whether some patients (such as women) were more likely to live alone or live without social support might need longer hospital LOS.

## 5. Conclusions

LOS has been reduced by almost 40% over the last 13 years, which poses a major cost saving to the public health system. Women, Māori, Pacific and Asian patients, older patients, people with more comorbidities and patients in earlier years were more likely to have extended LOS. Patients dispensed opiates on discharge had a longer LOS. Finally we found there were great geographical variations in LOS for primary hip and knee surgeries in New Zealand, which was likely to be multifactorial. LOS might reduce further with outpatient surgeries being possible and enhancement in the current recovery and rehabilitation programs.

## Figures and Tables

**Table 1 ijerph-16-04789-t001:** Length of hospital stay by subgroup.

Subgroup	Primary Hip Replacement					Primary Knee Replacement				
	Number of Admissions	LOS				Number of Admissions	LOS			
		Mean	Median	SD	*p*-Value		Mean	Median	SD	*p*-Value
**Gender**										
Female	29,015 (54.3%)	5.3	5	3.3	<0.001	26,698 (53.3%)	5.4	5	2.6	<0.001
Male	24,424 (45.7%)	4.7	4	3.0		23,374 (46.7%)	5.1	4	3.0	
**Ethnicity**										
European/others	46,347 (86.7%)	5.0	4	3.2	<0.001	41,799 (83.5%)	5.3	5	2.8	<0.001
Māori	6071 (11.4%)	4.8	4	3.1		3889 (7.8%)	5.1	5	2.8	
Pacific	698 (1.3%)	5.0	4	3.5		2477 (4.9%)	5.4	5	3.2	
Asian	323 (0.6%)	5.3	4	3.7		1907 (3.8%)	5.4	5	2.5	
**Age**										
<40	771 (1.4%)	4.3	4	2.7	<0.001	72 (0.1%)	5.1	5	3.0	<0.001
40–49	3126 (5.8%)	4.2	4	3.0		1130 (2.3%)	4.7	4	2.5	
50–59	8733 (16.3%)	4.3	4	2.4		7718 (15.4%)	4.7	4	2.3	
60–69	15,830 (29.6%)	4.6	4	2.6		16,973 (33.9%)	4.9	4	2.4	
70–79	17,645 (33.0%)	5.2	5	3.0		17,969 (35.9%)	5.4	5	2.9	
80+	7341 (13.7%)	6.7	6	4.7		6210 (12.4%)	6.5	6	3.5	
**Charlson comorbidity index**										
0	46,789 (87.6%)	4.8	4	2.8	<0.001	42,603 (85.1%)	5.1	5	2.4	<0.001
1	3504 (6.6%)	6.1	5	4.1		3781 (7.6%)	6.1	5	4.0	
2	2213 (4.1%)	6.3	5	4.8		2678 (5.3%)	6.2	5	3.7	
3+	933 (1.7%)	8.3	7	6.9		1010 (2.0%)	7.9	7	5.2	
**Use of opiate post-op**										
No	41,500 (77.7%)	4.9	4	3.0	<0.001	34,820 (69.5%)	5.2	5	2.7	<0.001
Yes	11,939 (22.3%)	5.6	4	3.8		15,252 (30.5%)	5.6	5	2.9	
**Year**										
2005	3283 (6.1%)	6.2	6	3.2	<0.001	3093 (6.2%)	6.6	6	3.5	<0.001
2006	3400 (6.4%)	6.1	6	3.7		3155 (6.3%)	6.4	6	2.9	
2007	3857 (7.2%)	6.1	5	3.3		3573 (7.1%)	6.2	6	3.0	
2008	3725 (7.0%)	5.9	5	3.8		3347 (6.9%)	6.1	5	3.0	
2009	3946 (7.4%)	5.6	5	3.1		3538 (7.1%)	5.9	5	2.8	
2010	3853 (7.2%)	5.5	5	3.6		3438 (6.9%)	5.8	5	2.9	
2011	3855 (7.2%)	5.2	5	2.7		3623 (7.2%)	5.5	5	2.7	
2012	4000 (7.5%)	4.9	4	3.1		3818 (7.6%)	5.3	5	2.7	
2013	4188 (7.8%)	4.6	4	2.9		3750 (7.5%)	5.0	5	2.6	
2014	4653 (8.7%)	4.4	4	2.9		4481 (8.9%)	4.6	4	2.5	
2015	4626 (8.7%)	4.2	4	2.8		4349 (8.7%)	4.3	4	2.1	
2016	4781 (8.9%)	4.0	3	2.9		4644 (9.3%)	4.2	4	2.2	
2017	5272 (9.9%)	3.8	3	2.4		5263 (10.5%)	4.1	4	2.1	
**Overall**	53,439	5.0	4	3.2		50,072	5.3	5	2.8	

**Table 2 ijerph-16-04789-t002:** Length of hospital stay by ethnicity after stratifying by age group.

Age	Ethnicity		Primary Hip Replacement				Primary Knee Replacement		
		Number of Admissions	LOS			Number of Admissions	LOS		
			Mean	Median	SD		Mean	Median	SD
**<40**									
	European/others	524	4.1	4	2.3	52	5.3	5	3.2
	Māori	169	4.3	4	3.5	9	5.7	5	2.9
	Pacific	64	5.0	4	3.7	9	3.8	3	2.4
	Asian	14	5.1	5	2.3	2	4.5	4.5	2.1
**40–49**									
	European/others	2156	4.2	4	3.0	817	4.7	4	2.6
	Māori	784	4.3	4	3.1	191	4.7	4	2.4
	Pacific	138	4.0	4	1.9	95	4.6	4	1.7
	Asian	48	4.3	4	2.0	27	5.6	5	2.8
**50–59**									
	European/others	6721	4.2	4	2.2	5740	4.6	4	2.0
	Māori	1740	4.3	4	2.8	1069	4.7	4	2.8
	Pacific	208	4.7	4	3.6	623	5.0	4	3.7
	Asian	63	5.4	5	5.0	286	5.3	5	2.5
**60–69**									
	European/others	13,597	4.6	4	2.6	13,600	4.8	4	2.3
	Māori	1997	4.7	4	3.0	1533	5.0	4	2.7
	Pacific	149	5.4	5	3.9	1045	5.3	5	2.5
	Asian	86	5.0	4	2.8	795	5.3	5	2.3
**70–79**									
	European/others	16,237	5.2	5	3.0	15,723	5.4	5	2.9
	Māori	1201	5.5	5	3.2	937	5.6	5	2.8
	Pacific	118	5.9	5	3.6	636	6.0	5	3.8
	Asian	86	5.8	5	3.7	673	5.5	5	2.5
**80+**									
	European/others	7112	6.7	6	4.7	5867	6.5	6	3.5
	Māori	180	6.4	6	4.4	150	6.4	6	3.1
	Pacific	21	6.8	6	4.5	69	6.6	5	3.7
	Asian	26	7.0	5	5.0	124	6.2	5	2.9

**Table 3 ijerph-16-04789-t003:** Length of hospital stay by DHB.

DHB		Primary Hip Replacement				Primary Knee Replacement		
	Number of Admissions	LOS			Number of Admissions	LOS		
		Mean	Median	SD		Mean	Median	SD
Northland	2521	5.1	5	3.4	2396	5.6	5	2.9
Waitemata	5021	4.7	4	2.8	600	5.0	4	3.4
Auckland	2630	5.6	5	4.9	3331	6.0	5	3.2
Counties Manukau	4064	4.9	4	3.1	5395	5.3	5	3.2
Waikato	4965	5.5	5	3.5	4731	5.6	5	2.9
Lakes	1581	4.7	4	2.7	1449	4.9	4	2.6
Bay of Plenty	3834	5.3	5	3.6	3375	5.4	5	3.0
Tairāwhiti	786	5.9	5	4.5	572	5.7	6	2.4
Hawkes Bay	2338	5.1	5	3.1	2050	5.1	5	2.5
Taranaki	1725	5.3	5	3.7	1202	5.6	5	2.9
Mid Central	2390	4.9	4	2.5	2275	5.0	4	2.2
Whanganui	1357	5.1	4	2.8	1264	4.9	4	2.5
Capital and Coast	2561	4.6	4	3.1	2696	4.8	4	2.3
Hutt Valley	1539	4.9	5	2.6	1551	5.1	5	2.5
Wairarapa	764	5.2	5	2.8	666	5.9	5	2.9
Nelson–Marlborough	2655	4.4	4	2.3	2108	4.4	4	2.0
West Coast	709	5.1	5	2.3	665	5.4	5	2.3
Canterbury	6081	4.8	4	2.8	4255	5.0	5	2.1
South Canterbury	1327	4.3	4	2.3	999	4.8	4	2.7
Southern	4558	5.2	5	2.8	3068	5.6	5	2.6
Unknown	33	5.8	5	3.3	24	5.3	5	1.6
Overall	53,439	5.0	4	3.2	5072	5.3	5	2.8

**Table 4 ijerph-16-04789-t004:** Adjusted odds ratio of length of stay over 5 days and 7 days after hip replacement surgeries. ^†^.

Subgroup	Over 5 Days				Over 7 Days			
	*p*-Value	Odds Ratio (95% CI)			*p*-Value	Odds Ratio (95% CI)		
**Gender**								
Female	Ref				Ref			
Male	<0.001	0.67	0.64	0.70	<0.001	0.81	0.76	0.86
**Ethnicity**								
European/others	Ref				Ref			
Māori	0.014	1.10	1.02	1.18	0.002	1.18	1.06	1.31
Pacific	<0.001	2.00	1.64	2.43	<0.001	2.06	1.60	2.67
Asian	0.002	1.54	1.17	2.02	0.026	1.50	1.05	2.15
**Age**	<0.001	1.06	1.06	1.07	<0.001	1.07	1.06	1.07
**Charlson Comorbidity Index**								
0	Ref				Ref			
1	<0.001	1.95	1.80	2.11	<0.001	2.30	2.09	2.53
2	<0.001	2.27	2.06	2.50	<0.001	3.03	2.70	3.39
3+	<0.001	4.42	3.79	5.16	<0.001	5.23	4.51	6.07
**Use of opiate post operation**								
No	Ref				Ref			
Yes	<0.001	1.70	1.62	1.79	<0.001	1.68	1.57	1.80
**Year** (Continuous)	<0.001	0.80	0.79	0.80	<0.001	0.84	0.84	0.85
**DHB**								
Northland	<0.001	2.33	2.07	2.62	0.001	1.30	1.11	1.53
Waitemata	Ref				Ref			
Auckland	<0.001	1.99	1.77	2.24	<0.001	1.66	1.43	1.92
Counties Manukau	0.107	1.09	0.98	1.22	0.900	0.99	0.86	1.14
Waikato	<0.001	2.61	2.36	2.88	<0.001	1.50	1.31	1.70
Lakes	<0.001	1.61	1.40	1.85	0.288	1.11	0.91	1.35
Bay of Plenty	<0.001	2.26	2.04	2.51	0.017	1.19	1.03	1.37
Tairāwhiti	<0.001	4.88	4.09	5.81	<0.001	3.13	2.53	3.86
Hawkes Bay	0.068	1.13	0.99	1.28	<0.001	0.70	0.58	0.84
Taranaki	<0.001	1.75	1.53	2.00	0.003	1.31	1.09	1.56
Mid Central	<0.001	1.37	1.21	1.55	0.015	0.80	0.66	0.96
Whanganui	<0.001	1.59	1.37	1.85	0.113	0.84	0.67	1.04
Capital and Coast	0.031	0.87	0.76	0.99	<0.001	0.59	0.49	0.71
Hutt Valley	<0.001	1.54	1.34	1.78	0.024	0.78	0.63	0.97
Wairarapa	<0.001	2.54	2.12	3.04	0.001	1.50	1.17	1.91
Nelson–Marlborough	0.026	0.87	0.76	0.98	<0.001	0.54	0.44	0.65
West Coast	<0.001	2.33	1.93	2.81	0.182	1.20	0.92	1.58
Canterbury	0.181	1.07	0.97	1.18	<0.001	0.52	0.45	0.61
South Canterbury	<0.001	0.75	0.63	0.88	<0.001	0.46	0.35	0.61
Southern	<0.001	1.78	1.61	1.97	0.002	1.23	1.08	1.41
Unknown	0.021	2.64	1.16	6.04	0.295	1.73	0.62	4.85

^†^ Adjusted for age, gender, ethnicity, Charlson comorbidity index score, year of surgery, use of opiate or not after surgery and DHB.

**Table 5 ijerph-16-04789-t005:** Adjusted odds ratio of length of stay over 5 days and 7 days after knee replacement surgeries. ^†^.

Subgroup	Over 5 Days				Over 7 Days			
	*p*-Value	Odds Ratio (95% CI)			*p*-Value	Odds Ratio (95% CI)		
**Gender**								
Female	Ref				Ref			
Male	<0.001	0.76	0.72	0.79	<0.001	0.90	0.85	0.95
**Ethnicity**								
European/others	Ref				Ref			
Māori	0.006	1.12	1.03	1.22	0.016	1.15	1.03	1.28
Pacific	<0.001	1.38	1.25	1.52	0.007	1.20	1.05	1.38
Asian	<0.001	1.35	1.21	1.51	0.068	1.15	0.99	1.33
**Age**	<0.001	1.05	1.05	1.05	<0.001	1.05	1.05	1.06
**Charlson Comorbidity Index**								
0	Ref				Ref			
1	<0.001	1.71	1.59	1.85	<0.001	2.02	1.85	2.22
2	<0.001	2.04	1.87	2.22	<0.001	2.34	2.11	2.60
3+	<0.001	3.98	3.44	4.61	<0.001	5.06	4.38	5.83
**Use of opiate post operation**								
No	Ref				Ref			
Yes	<0.001	1.65	1.58	1.73	<0.001	1.76	1.65	1.88
**Year** (Continuous)	<0.001	0.80	0.79	0.80	<0.001	0.83	0.83	0.84
**DHB**								
Northland	<0.001	1.98	1.78	2.21	<0.001	1.42	1.23	1.64
Waitemata	Ref				Ref			
Auckland	<0.001	2.06	1.87	2.28	<0.001	1.79	1.58	2.03
Counties Manukau	0.474	1.03	0.95	1.13	0.849	1.01	0.90	1.14
Waikato	<0.001	2.02	1.85	2.21	<0.001	1.54	1.37	1.73
Lakes	0.025	1.17	1.02	1.34	0.430	1.08	0.89	1.30
Bay of Plenty	<0.001	1.55	1.40	1.71	0.108	1.12	0.98	1.28
Tairāwhiti	<0.001	2.78	2.30	3.37	0.006	1.43	1.11	1.85
Hawkes Bay	0.008	0.85	0.75	0.96	<0.001	0.56	0.47	0.68
Taranaki	<0.001	1.96	1.70	2.27	<0.001	1.59	1.33	1.90
Mid Central	0.882	1.01	0.90	1.14	<0.001	0.65	0.54	0.79
Whanganui	0.431	1.06	0.91	1.23	0.002	0.69	0.55	0.87
Capital and Coast	0.017	0.87	0.78	0.98	0.005	0.80	0.68	0.93
Hutt Valley	0.020	1.17	1.03	1.34	0.282	0.90	0.75	1.09
Wairarapa	<0.001	2.41	2.01	2.88	<0.001	1.72	1.38	2.16
Nelson–Marlborough	<0.001	0.56	0.49	0.64	<0.001	0.45	0.36	0.55
West Coast	<0.001	2.07	1.73	2.48	0.266	1.16	0.89	1.49
Canterbury	0.053	0.91	0.82	1.00	<0.001	0.57	0.49	0.66
South Canterbury	<0.001	0.73	0.61	0.86	0.027	0.77	0.61	0.97
Southern	<0.001	1.61	1.46	1.79	<0.001	1.29	1.13	1.47
Unknown	0.103	2.16	0.86	5.47	0.585	0.65	0.14	3.06

^†^ Adjusted for age, gender, ethnicity, Charlson comorbidity index score, year of surgery, use of opiate or not after surgery and DHB.

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
