# Peer review of "Length of Hospital Stay for Osteoarthritic Primary Hip and Knee Replacement Surgeries in New Zealand"

_ijerph, 2019, doi:10.3390/ijerph16234789_

Round 1

Reviewer 1 Report

This paper could be accepted as a preliminary study to assess hospitalizatiion during osteoarthritic primary hip and knee replacement surgeries in New Zealand.

If the study is supported by funds it would be necessary to evaluate the differences in age and clinical condition of the patient.
Furthermore, morphological analysis of the tissues must be carried out.
Furthermore it is necessary to plan a preventive and post-operative activity and therapy to reduce hospital stay and costs.

Author Response

This paper could be accepted as a preliminary study to assess hospitalizatiion during osteoarthritic primary hip and knee replacement surgeries in New Zealand.

If the study is supported by funds it would be necessary to evaluate the differences in age and clinical condition of the patient.

Furthermore, morphological analysis of the tissues must be carried out.

Response: The differences in LOS in age and clinical condition (Charlson Comorbidity Index score) have been examined. Multivariate analysis has been carried out to examine the impact of relevant factors on LOS.

Furthermore it is necessary to plan a preventive and post-operative activity and therapy to reduce hospital stay and costs.

Response: As mentioned in the second and third paragraph in the discussion, protocols with an emphasis on recovery and rehabilitation have been introduced in elective orthopaedics in New Zealand, including ‘Accelerated Rehabilitation’, ‘Fast-Track’, ‘Clinical Pathways’ and “Enhanced Recovery After Surgery (ERAS)” programmes. These programmes have been shown to be efficient in reducing LOS. The LOS could be reduced further, with outpatient hip and knee replacement surgeries being possible. A US study found that 1-day LOS discharge after total hip and knee replacement is achievable and did not increase readmissions compared to 2-day LOS discharge.

Reviewer 2 Report

As a practicing surgeon, I've lived through the decrease in LOS you've described nicely in the past 10 years.  Some determinants of discharge times were not discussed in your paper, and may confound your results and make the conclusions different. Obviously, some of this data may not be available in the database you accessed - this would be important to note as limitations.

Skilled nursing facilities or extended rehabilitation facilities: if they were to be present in some jurisdictions, but not others, it would bias the data. Social determinants: were women more likely to live alone than men?  Were people in some jurisdictions more likely to live without social support?

Why not analyze the data by year also?  Could it be that some very long LOS in the early part of the cohort biased the data?  Are the jurisdictions disparate now in terms of LOS?  If you report mean, SD and median, it may describe the data more clearly.

Author Response

As a practicing surgeon, I've lived through the decrease in LOS you've described nicely in the past 10 years.  Some determinants of discharge times were not discussed in your paper, and may confound your results and make the conclusions different. Obviously, some of this data may not be available in the database you accessed - this would be important to note as limitations.

Skilled nursing facilities or extended rehabilitation facilities: if they were to be present in some jurisdictions, but not others, it would bias the data. Social determinants: were women more likely to live alone than men?  Were people in some jurisdictions more likely to live without social support?

Response: Thank you for your recommendations. We have added these in the study limitations (last paragraph of the discussion).

Why not analyze the data by year also?  Could it be that some very long LOS in the early part of the cohort biased the data?  Are the jurisdictions disparate now in terms of LOS?  If you report mean, SD and median, it may describe the data more clearly.

Response: Year was included in all the analyses. The odds ratio of other factors (Table 3 and Table 4) in the multivariate analyses were adjusted for year and others. Median and SD of LOS have been added in Table 1.

Reviewer 3 Report

Brief Overview
This manuscript summarizes over 100,000 osteoarthritic hip and knee replacements in New Zealand in 2005-2017 and compares the length of stay (LOS) by gender, ethnicity, age, comorbidity score, year of surgery, use of opioids, and district. The authors found that LOS has reduced by 40% over the last 13 year and that female, Maori, Pacific, and Asian patients, older patients, people with comorbidities or opioids are more likely to have extended LOS.

Major Comments
1. This manuscript does not provide impactful conclusions. It has been well accepted that female patients (Rissanen 1996, Liebergall 1999, Watkins 1999, Hayes 2000, Husted 2008), older patients (Kim 1993, Wang 1998, Watkins 1999, Hayes 2000, Husted 2008), and patients with comorbidities (Husted 2008) or opioids (Oderda 2013) have longer LOS after joint replacements. It will be useful if the authors include rehabilitation after surgery (Roos 2003), pre-operative education (Jones 2011), early mobilization (Guerra 2015), or any other information to strengthen the impact of this manuscript.
2. The authors say Maori, Pacific, and Asian patients were more likely to have extended LOS in the Abstract and Conclusions. This is inconsistent with line 96 or Table 1.

Minor Comments
1. Please review the reference formatting in the Instructions for Authors. In the text, reference numbers should be placed in square brackets [ ], and placed before the punctuation; for example [1], [1–3] or [1,3].

Author Response

Brief Overview

This manuscript summarizes over 100,000 osteoarthritic hip and knee replacements in New Zealand in 2005-2017 and compares the length of stay (LOS) by gender, ethnicity, age, comorbidity score, year of surgery, use of opioids, and district. The authors found that LOS has reduced by 40% over the last 13 year and that female, Maori, Pacific, and Asian patients, older patients, people with comorbidities or opioids are more likely to have extended LOS. 

Major Comments

This manuscript does not provide impactful conclusions. It has been well accepted that female patients (Rissanen 1996, Liebergall 1999, Watkins 1999, Hayes 2000, Husted 2008), older patients (Kim 1993, Wang 1998, Watkins 1999, Hayes 2000, Husted 2008), and patients with comorbidities (Husted 2008) or opioids (Oderda 2013) have longer LOS after joint replacements. It will be useful if the authors include rehabilitation after surgery (Roos 2003), pre-operative education (Jones 2011), early mobilization (Guerra 2015), or any other information to strengthen the impact of this manuscript. 

Response: We included acknowledgement of the importance of the recovery and rehabilitation programmes in the third paragraph in the discussion. We have also added more information saying further reduction in LOS is possible with reference to the day surgery for hip and knee replacement in the US.

The authors say Maori, Pacific, and Asian patients were more likely to have extended LOS in the Abstract and Conclusions. This is inconsistent with line 96 or Table 1. 

Response: Before adjustment for age and other factors, Māori have a shorter average LOS than NZ European/others (Table 1). Māori patients are more likely to be younger and younger patients tended to have a shorter LOS. Therefore, after adjustment for age and other factors, Māori were more likely to have extended LOS (more than 5 days and more than 7 days in Table 3 and 4).

Minor Comments

Please review the reference formatting in the Instructions for Authors. In the text, reference numbers should be placed in square brackets [ ], and placed before the punctuation; for example [1], [1–3] or [1,3].

Response: the reference format has been changed.

Reviewer 4 Report

Authors could explore the reasons for the effect of gender and age on LOS in more depth in the discussion section since that information will be clinically relevant. 

Authors could also comment and provide recommendations on how LOS could be reduced further and made comparable with US and UK data. 

Author Response

Authors could explore the reasons for the effect of gender and age on LOS in more depth in the discussion section since that information will be clinically relevant. 

Response: We have added some comments in the fifth paragraph in the discussion: “In our study, women have a longer LOS than men, which is consistent with other studies [6, 26, 27]. This could be attributed to differences in ways in which men and women respond to the disease, anesthesia and the surgery or to bias on the part of healthcare workers [6, 26, 27]. As expected, the LOS increased with age, because older patients have more comorbidities and have slower recovery after surgery. It has also been found that older patients were more likely to have experience post-operative complications, admission to the ICU, and be discharged to a skilled care facility [28].”

Authors could also comment and provide recommendations on how LOS could be reduced further and made comparable with US and UK data. 

Response: The LOS in NZ is shorter than the LOS in the UK but longer than the LOS in the US. I have commented on how the LOS could be reduced further in the second paragraph in the discussion and the conclusion “LOS may drop further with outpatient surgeries becoming possible and enhancement in the current recovery and rehabilitation programmes.”

Round 2

Reviewer 3 Report

In this revised manuscript, one of the two major comments have been addressed by the authors with additional paragraphs in Discussion. However, the reviewer is still concerned for the statement “Maori, Pacific, and Asian patients were more likely to have extended LOS.” 

In the original submission, authors said Maori, Pacific, and Asian patients were more likely to have extended LOS in the Abstract and Conclusions which were inconsistent with line 96 and Table 1. The authors responded in this revision, “Before adjustment for age and other factors, Māori have a shorter average LOS than NZ European/others (Table 1). Māori patients are more likely to be younger and younger patients tended to have a shorter LOS. Therefore, after adjustment for age and other factors, Māori were more likely to have extended LOS (more than 5 days and more than 7 days in Table 3 and 4).”

In Table 1, the mean LOS after hip replacement is 4.8, 5.0, 5.3, and 5.0 for Maori, Pacific, Asian, and European patients, respectively. In Table 3, was the LOS adjusted based on the mean LOS in Table 1? What were the adjusted LOS values? Is it likely to increase from 4.8 to >5.0 for Maori patients and from 5.0 to >5.0 for Pacific patients after age adjustment? It will be beneficial to explain this adjustment in Methods (line 98-100) and in Results (line 128-131). Instead of stating Table 3 has a similar pattern as Table 1, authors should explain there is an age adjustment so the Maori and Pacific LOS changed from shorter LOS to longer LOS when compared to that of European patients. The authors should also emphasize that the LOS values go into the calculation of odds ratios are adjusted by age in Table 3. Maybe add a note or asterisk in the Table legend. Similarly, for knee replacement, authors should explain why the original LOS of Maori patients in Table 1 suggesting a shorter LOS is different from the odds ratio for Maori in Table 4 suggesting the opposite. 

It will be beneficial for authors to add the above information.

Author Response

Thank you for your recommendations. To explain the changes from Table 1 to Table 3 and Table 4, we have added a Table (new Table 2) showing the mean LOS by ethnicity after stratifying them by age group. We can see from the new Table 2 that in most age groups, Maori and Pacific patients had a longer LOS than European/others. Because Maori and Pacific patients were more likely to be in the younger age groups which have a shorter LOS, the average LOS for all age groups together was shorter for Maori and Pacific. I have also added some texts in the methods (line 81), results (line 98, and line 113) and tables (notes in new Table 4 and 5) saying that the odds ratios were adjusted for age and other factors. Hope these would clarify the confusion.